# Secondary Succession in the Tropical Lowland Rainforest Reduced the Stochasticity of Soil Bacterial Communities through the Stability of Plant Communities

Xuan Hu [1,2], Qi Shu [3], Zean Shang [4], Wen Guo [1] and Lianghua Qi [1,*]

[1] Key Laboratory of National Forestry and Grassland Administration/Beijing for Bamboo & Rattan Science and Technology, International Centre for Bamboo and Rattan, Beijing 100102, China; huxuan@icbr.ac.cn (X.H.); guowen@icbr.ac.cn (W.G.)

[2] National Positioning and Monitoring Station for Ecosystem of Bamboo and Rattan, Sanya 572000, China

[3] Research Institute of Forestry, Chinese Academy of Forestry, Beijing 100091, China; shuqi1005@caf.ac.cn

[4] Shaanxi Academy of Forestry, Xi'an 710082, China; sza1001@sina.com

[*] Correspondence: qlh@icbr.ac.cn; Tel.: +86-13552937880

**Abstract:** The effects of natural succession on plant and soil bacterial communities were previously established, but changes in plant and soil bacterial communities and their response to soil properties are not well characterized in different stages of secondary forest succession, especially in tropical regions with endemic plant species. We investigated the dynamics of plant communities, soil properties and the structure of soil bacterial communities at sites representing 33 (early successional stage), 60 (early-mid successional stage) and 73 (mid successional stage) years of secondary succession in the tropical lowland rainforest of Hainan, China, by using 16S rRNA high-throughput sequencing. From the perspective of plant composition, the number of families, genera and species were increasing along with the progress of succession. Additionally, the changes in the ranking of important values along with the progress of the forest succession were consistent with the niche width calculated by the previous stage of the plant community. The results of niche overlap, Pearson's correlation and Spearman's rank correlation coefficients and significance indicated that in the early stage of succession, tree species did not fully utilize environmental resources. Then, as time went by, the number of negative correlations of plants in the early-mid stage was more than that in the mid stage of succession. Significant differences were found in the species richness of soil microorganisms among the three successional stages. Nutrient contents in early successional stage rainforests were less abundant than in early-mid and mid forest soils. The influence of soil nutrient concentration, particularly N and P content, on soil bacterial composition at the phylum level was larger in the early-mid stage than in the mid stage. The stochasticity of the soil bacterial community at the early successional stage of the rainforest was significantly higher than that at mid stage. Overall, as the diversity of plant communities increased, the competition decreased, the soil nutrient content changed and the stochasticity of soil bacterial communities decreased as a result of forest succession.

**Keywords:** tropical lowland rainforest; successional stage; plant communities composition; soil bacterial community; stochasticity

## 1. Introduction

Tropical forests are one of the most important biomes worldwide. With only 12% of the global land area, they act as both global warehouses of biodiversity (containing an estimated 50% of species) and of carbon (C) by storing 25% of plant biomass [1]. Tropical forests also dominate terrestrial C dynamics, contributing almost 70% of the global gross forest C sink for the time period of 1990–2007 [2]. Less than 50% of the world's tropical forests remain standing today, with much of the remaining forest cover seriously affected by human activities, such as logging, fires, fragmentation, mining, hunting, and conversion to pastures. The degradation

of tropical rain forests may have strong negative consequences for biodiversity, climate normalization, and the well-being of rural and urban populations [3,4].

It was found that natural plant secondary succession without human disturbance is an effective way to improve soil conditions and restore damaged environments [5]. Vegetation succession is the process of restoring plant and soil communities [6,7] and is one of the strategies used to control land degradation [8]. Several studies were conducted to learn how the plant and soil bacterial community changes during secondary forest succession [9]; however, our understanding of secondary processes in the tropical regions where plant and soil bacterial species diversity and composition usually change rapidly is still poor, especially under the situation that few studies can conduct a comprehensive survey of tropical rain forest plants.

Hainan Island is located on the northern edge of the tropical zone. It is the tropical zone with the largest area of island-type tropical rainforests and the greatest species diversity in China. Hainan Island had multiple separations and reunions with the mainland throughout the history of the earth's crust. Due to its unique geographical location, this place acted as a biological refuge during the ice period, preserving a diverse range of biological species. The composition and distribution of native plant species are distinctive; for example, Hainan's endemic plant species, *Hopea exalata*, is found exclusively in this area and in a wide range of habitats [10]. The information on the relationship between plant, soil variables and soil bacterial communities in places with endemic plant species is still scarce. This knowledge is essential for understanding the progress of secondary forest succession and for the conservation of the ecological environment.

The present study investigated the dynamics of plant communities, soil properties and the structure of soil bacterial communities at sites representing 33, 60 and 73 years of secondary succession in the tropical lowland rainforest of Hainan, China, by using 16S rRNA high-throughput sequencing and investigating forest plants. Our objective was to (1) evaluate the patterns of change in soil properties, plant communities and soil bacterial communities in three different successional forest stages, (2) to investigate if changes between the plant and soil bacterial communities are congruous, (3) determine the response of soil bacterial communities–soil properties–plant community along the chronosequence.

## 2. Materials and Methods

### 2.1. Study Site Description

The study site is located in the Ganshiling Nature Reserve (2103.44 ha) of Hainan, China (109°34′ E–109°42′ E, 18°20′ N–18°21′ N). The vegetation type is tropical lowland rainforest. This zone is an area of tropical marine monsoon climate It has distinct dry and rainy (May to October) seasons. The average annual rainfall is 1800 mm, and the mean annual temperature is 24.5 °C [11]. The altitude of the area is 50–681 m, and soils have formed by the weathering of granite. These soils are classified as red soil according to the Chinese soil classification system, which is equivalent to an Oxisol in the USDA Soil Taxonomy [12].

For our study, a natural successional chronosequence of the rainforest communities was selected from this nature reserve, including early successional (33 years, Ganshiling Nature Reserve was established in 1985, there were no logging activities since then) tropical lowland rainforests, early-mid successional (about 60 years, 1958 People's commune movement needed to cut down a lot of wood) tropical lowland rainforests and mid successional (73 years, 1945 Second Sino-Japanese War ended, during the war, a lot of trees were harvested as wood) tropical lowland rainforests [13–15]. The representative species of early successional stage rainforests are *Aporosa dioica*, *Melastoma sanguineum* and *Phyllanthus emblica*. The representative species of early-mid successional stage rainforests are *Hopea exalata*, *Vatica mangachapoi* and *Koilodepas bainanense*. The dominant species of mid successional stage rainforests are *H. exalata*, *Olea dioica* and *Garcinia oblongifolia*.

*2.2. Experimental Design, Plant Community Surveying and Soil Sampling*

In June of 2018, a total of 30 plots (20 × 20 m), 10 each for the three kinds of successional stage rainforests, were established and used for soil sampling and plant community surveying. All the plots were on a similar slope position, with the elevation ranging from 90 to 340 m above sea level. To ensure spatial heterogeneity, each of the 30 plots (20 × 20 m) were separated from each other by at least 500 m.

Arbor species thrive in the Ganshiling area of Hainan Island, with plots dominated by tall trees with high canopy density and sparse understory vegetation. The tree species in the sample plots were identified, and each tree was examined. The diameter ($\geq$3 cm) at breast height was measured (tape measure) and recorded; tree height, crown width and height under branches were surveyed (Vertex Laser Ultrasonic Tree Height Range Finder, Haglof, Sweden).

A total of 30 samples (3 successional stages × 10 plots) were collected. In each plot, after removing the litter layer, 3–5 soil samples of each of 0–10 cm, 10–30 cm, 30–50 cm depth were collected in an S-shaped pattern, incorporating spatial heterogeneity at the plot scale, and were mixed to form a final composite sample [16,17], and kept on ice. All the soil samples were passed through a 2 mm sieve to remove debris such as large roots and rocks. Then, each sample was divided into two parts: one was stored at −80 °C for DNA extraction, the other was stored at 4 °C for soil property measurements.

*2.3. Analysis of Soil Properties*

Soil pH was measured in the soil–water slurry (1:5) using a pH meter. Total organic carbon (TOC) and total nitrogen (TN) of soil were measured by an elemental analyzer (Costech ECS 4024 CHNSO, Picarro, Italy). Soil total phosphorus (TP) was determined by an automatic chemical analyzer (Smartchem 300, AMS, Italy) with a Mo-Sb colorimetric method. Soil total potassium (TK) was measured by a flame photometer (M410, Sherwood, UK). Atomic absorption spectrophotometry was employed to measure soil available potassium (AK). Soil available phosphorus (AP) was determined with the ultraviolet spectrophotometry method. Alkali nitrogen (AN) was measured by a Kjeldahl method analyzer (Kjeldahl 2300, Foss, Denmark).

*2.4. Soil Samples DNA Extraction, PCR Amplification, and Sequencing*

2.4.1. DNA Extraction

Genomic DNA was extracted using a MOBIO Power Soil® DNA Isolation Kit (MOBIO Laboratories, Carlsbad, CA, USA) in accordance with the manufacturer's instructions. The concentration and purity were measured using the NanoDrop One (Thermo Fisher Scientific, Waltham, MA, USA).

2.4.2. PCR Amplification

16S rRNA genes of distinct regions V4–V5 were amplified using specific primer 515F and 907R with 12bp barcode. Primers were synthesized by Invitrogen (Invitrogen, Carlsbad, Carlsbad, CA, USA). PCR reactions, containing 25 µL 2× Premix Taq (Takara Biotechnology, Dalian Co. Ltd., Dalian, China), 1 µL of each primer (10 mM) and 3 µL DNA (20 ng/µL) template in a volume of 50 µL, were amplified by thermocycling: 5 min at 94 °C for initialization; 30 cycles of 30 s denaturation at 94 °C, 30 s annealing at 52 °C and 30 s extension at 72 °C; followed by 10 min final elongation at 72 °C. The PCR instrument was BioRad S1000 (Bio-Rad Laboratory, Hercules, CA, USA).

The length and concentration of the PCR product were detected by 1% agarose gel electrophoresis. Samples with a bright main strip between 400–450 bp were used for further experiments. PCR products were mixed in equidensity ratios according to the GeneTools Analysis Software (Version4.03.05.0, SynGene, Cambridge, UK). Then, a mixture of PCR products was purified with the E.Z.N.A.® Gel Extraction Kit (Omega, Norcross, GA, USA). Sequencing libraries were generated using the NEBNext® Ultra™ DNA Library Prep Kit for Illumina® (New England Biolabs, Ipswich, MA, USA) following the manufacturer's

recommendations, and index codes were added. The library quality was assessed on the Qubit@ 2.0 Fluorometer (Thermo Fisher Scientific, Waltham, MA, USA) and Agilent Bioanalyzer 2100 system (Agilent Technologies, Waldbronn, Germany). At last, the library was sequenced on an IlluminaHiseq2500 platform, and 250 bp paired-end reads were generated (Guangdong Magigene Biotechnology Co. Ltd., Guangzhou, China).

2.4.3. Sequence Processing

Quality filtering on the paired-end raw reads was performed under specific filtering conditions to obtain the high-quality clean reads according to the Trimmomatic (V0.33) quality-controlled process. Paired-end clean reads were merged using FLASH (V1.2.11), and the spliced sequences were called raw tags. Sequences were assigned to each sample based on their unique barcode and primer using Mothur software (V1.35.1), after which the barcodes and primers were removed and the effective Clean Tags were obtained. Sequence analysis was performed using usearch software (V10). For each representative sequence, the silva database was used to annotate taxonomic information.

*2.5. Data Analysis*

Statistical analyses of the data were carried out using SPSS 11.5 for Windows (SPSS Inc, Chicago, IL, USA). The plant important value and Niche breadth were calculated using the following equations [18]:

$$IV_i = RD_i + RF_i + RP_i \tag{1}$$

$$RDi = \frac{n_i}{N} \times 100\% \tag{2}$$

$$RFi = \frac{F_i}{F} \times 100\% \tag{3}$$

$$RPi = \frac{G_i}{G} \times 100\% \tag{4}$$

where $IV_i$ is the important value, $RD_i$ is relative density, $n_i$ is the number of one species, $N$ is the number of all the plants. $RF_i$ is the relative frequency of one species, $F_i$ is the frequency of one species, $F$ is the sum of the frequencies of all species. $RP_i$ is relative coverage, $G_i$ is the area of one species, $G$ is the sum of the area of all species.

$$B_i = \frac{1}{\sum_{j=1}^{r} P_{ij}^2} \tag{5}$$

In this equation, $B_i$ is the niche breadth, $r$ means there are $r$ resource states in the resource matrix, $P_{ij} = N_{ij} / Y_i$ refers to the percentage of individuals of species $i$ in resource position $j$ in the total number of individuals of this species in the entire resource state.

The analysis of plant niche overlap and interspecific associations were performed using packages spaa, permute, lattice, vegan, psych and ggplot2 (R Core Team, 2019) written with the R language (https://cran.r-project.org/packages/spaa, accessed on 27 December 2021, permute, lattice, vegan, psych and ggplot2).

To explore the associations between soil bacterial community structure and the measured environmental factors, distance-based redundancy analysis (dbRDA) ordination plots were used to visualize the relationship [19]. dbRDA analysis was performed using a vegan package (R Core Team, 2019) written with the R language (https://cran.r-project.org/packages/vegan, accessed on 27 December 2021).

Normalized stochasticity ratio (NST), a new general mathematical framework to quantify ecological stochasticity under different situations, was used to reflect the contribution of stochastic assembly relative to deterministic assembly; this can serve as a better quantitative measure of stochasticity [20]. NST analysis was performed using an NST package (R Core Team, 2019) written with the R language (https://cran.r-project.org/packages/NST, accessed on 27 December 2021).

### 3. Results

#### 3.1. Environmental Factors Differences at Successional Stages

Nutrient contents in early successional stage rainforests were less abundant than in early-mid and mid forest soils (Table 1). To be more specific, except for the total phosphorus (TP) content, the content of all other nutrients was significantly lower in the early stage than in the early-mid and mid stages. The rainforest's soil nutrient content did not alter significantly between the early-mid and mid stages of succession, but in terms of quantity, all nutrients in the early-mid stage were greater than those in the latter stage, with the exception of TP and available phosphorus (AP). There was no significant difference in the pH value of the tropical rain forest soils among the three different successional stages, all showing an acidic state.

**Table 1.** Environmental factors differences at successional stages.

|  | Early | Early-Mid | Mid |
|---|---|---|---|
| Altitude (m) | 243.00 ± 8.17 [a] | 233.00 ± 23.11 [a] | 238.50 ± 29.19 [a] |
| pH | 4.69 ± 0.04 [a] | 4.74 ± 0.06 [a] | 4.79 ± 0.07 [a] |
| Organic matter (g/kg) | 10.54 ± 1.38 [b] | 24.27 ± 2.30 [a] | 19.29 ± 2.84 [a] |
| Total organic C (g/kg) | 6.11 ± 0.8 [b] | 14.08 ± 1.33 [a] | 11.19 ± 1.65 [a] |
| Total N (g/kg) | 0.45 ± 0.05 [b] | 1.19 ± 0.09 [a] | 0.98 ± 0.12 [a] |
| Total P (g/kg) | 0.12 ± 0.01 [a] | 0.14 ± 0.01 [a] | 0.18 ± 0.03 [a] |
| Alkali N (mg/kg) | 58.9 ± 7.03 [b] | 102.27 ± 9.16 [a] | 90.88 ± 13.62 [a] |
| Available P (mg/kg) | 0.11 ± 0.01 [b] | 0.53 ± 0.13 [a] | 0.54 ± 0.17 [a] |

Values are means ± standard error ($n = 10$). Significance is based on Duncan's multiple range test. Means followed by a common letter are not significantly different at the 5% level of significance.

#### 3.2. Plant Important Value and Niche Width

The classification and quantitative statistics of plant species in tropical rain forests at different successional stages (Table S1) showed that, from the perspective of plant composition, the number of families, genera and species was increasing along with the progress of succession. In terms of the number of species and species composition, the difference between the early stage, early-mid stage and mid stage was relatively large, and the early-mid stage and mid stage were relatively similar.

The important value is a comprehensive index used to indicate the status and role of a species in the community. The top five important values of plant communities in the early stage of succession were *Antidesma ghaesembilla*, *Melastoma sanguineum*, *Garcinia oblongifolia*, *Olea dioica* and *Schima superba* (Table S1a), most of which were sunny plant pioneer tree species. This phenomenon was in line with the characteristics of the plot being in the initial stage of restoration. The top five important values of plant communities in the early-mid stage of succession were *Hopea exalata*, *Vatica mangachapoi*, *Garcinia oblongifolia*, *Koilodepas bainanense* and *Schima superba* (Table S1b). Additionally, the top five of the mid stage were *H. exalata, V. mangachapoi, S. superba, G. oblongifolia* and *O. dioica* (Table S1c), in which there were three of the same species concerning the result for the early-mid stage. *H. exalata* was the plant with the highest important value in the early-mid and mid stages. This was in line with the fact that Ganshiling Nature Reserve is *H. exalata*'s nature reserve, and *H. exalata* was an endemic species in this region.

Niche represents the range of environmental conditions and resource quality within which an individual or species can survive and reproduce. Niche breadth Levins index calculation results showed that the top five tropical rain forest plant communities in the early successional stage were *Syzygium hainanense* (4.66), *A. ghaesembilla* (4.61), *Memecylon ligustrifolium* (4.05), *Phyllanthus emblica* (3.52) and *S. superba* (2.65). The top five plant communities in the early-mid successional stage were *V. mangachapoi* (5.77), *H. exalata* (5.57), *Dillenia turbinate* (4.86), *G. oblongifolia* (4.57) and *M. ligustrifolium* (4.33). Additionally, the top five plant communities in the mid successional stage were *G. oblongifolia* (5.62), *V. mangachapoi* (4.65), *H. exalata* (4.54), *S. superba* (3.72) and *Alphonsea monogyna* (3.66). The

niche width of the plant community was inconsistent with the order of importance values, indicating that the Ganshiling lowland rainforest community in Hainan Island was in the stage of development and restoration, and the community structure will undergo changes in a certain period of time in the future. This result is also in line with the definition that the forest is still in the middle of succession.

### 3.3. Plant Niche Overlap and Interspecific Associations

The darker the color in the niche overlap map, the greater the degree of niche overlap. In the early stage of tropical rainforest succession, there were a total of 946 pairs of plants, of which 348 pairs had niche overlap, accounting for 38.54% of the total, and 35 pairs with an overlap index of 1, accounting for 3.88% of the total (Figure 1a). In the early-mid stage, there were a total of 2926 pairs of plants, of which 1226 pairs had niche overlap, accounting for 41.90% of the total, and 70 pairs with an overlap index of 1, accounting for 2.39% of the total (Figure 1b). In the mid stage, there were a total of 3160 pairs of plants, of which 1187 pairs had niche overlap, accounting for 37.56% of the total, and 101 pairs with an overlap index of 1, accounting for 3.20% of the total (Figure 1c). In the early stage of succession, the niche overlap index of 1 accounted for the largest proportion, which means that in the early stage of restoration, tree species did not fully utilize environmental resources, the inter-species relationship was more inclined to inter-species competition and the community structure lacked a certain degree of stability. The environmental resources represent the probability of increasing biodiversity, and the community has a good potential for development and succession.

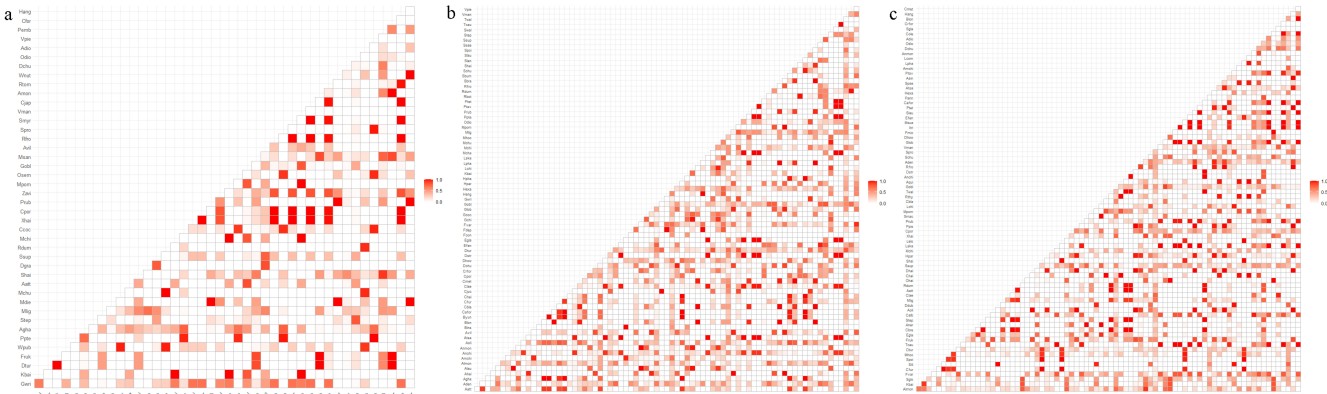

**Figure 1.** Pianka niche overlap index of tree species in different successional stages. In the figure, (**a**) represents forests of early successional stages, (**b**) represents forests of early-mid successional stages, (**c**) represents forests of mid successional stages.

The Pearson's correlation test result of the interspecific association (Figure 2) showed that there was no negative correlation between the tropical rainforest plant species in the early and mid stages of succession, and there were two pairs showing a negative correlation in the early-mid stage of succession; they were, *H. exalata–S. superba* and *H. exalata–Syzygium tephrodes*, respectively.

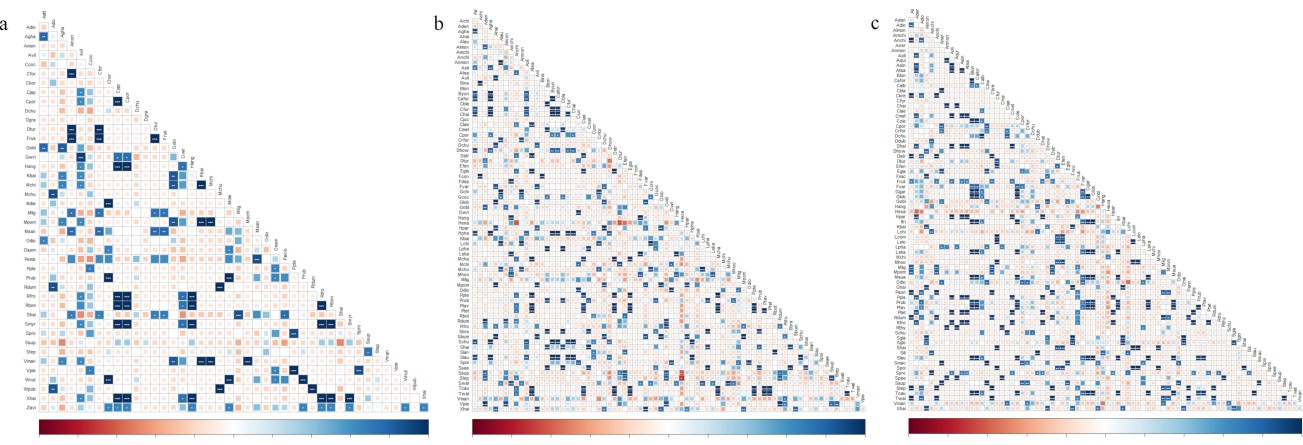

**Figure 2.** Pearson's correlation coefficients and significance of tree species in different successional stages. In the figure, (**a**) represents forests of early successional stages, (**b**) represents forests of early-mid successional stages, (**c**) represents forests of mid successional stages.

The Spearman's rank correlation coefficients and significance test result of the interspecific association (Figure 3) showed that there was no negative correlation between the tropical rainforest plant species in the early stages of succession, and there were six pairs showing a negative correlation in the early-mid and mid stages of succession. Among the six pairs of plant species that showed a negative correlation in the early-mid stage of succession, five pairs had $p < 0.05$ (*H. exalata-Rhodamnia dumetorum, H. exalata-D. turbinate, Diospyros howii-D. turbinate, D. howii-O. dioica, Croton laevigatus- Diospyros chunii*), and one pair had $p < 0.01$ (*H. exalata- Syzygium tephrodes*). All six pairs that showed a negative correlation in the mid stage of succession had $p < 0.05$ (*Heritiera angustata- A. monogyna, H. angustata- G. oblongifolia, H. angustata- V. mangachapoi, H. exalata- Ardisia densilepidotula, H. exalata- Phoebe tavoyana, S. superba- Litchi chinensis*).

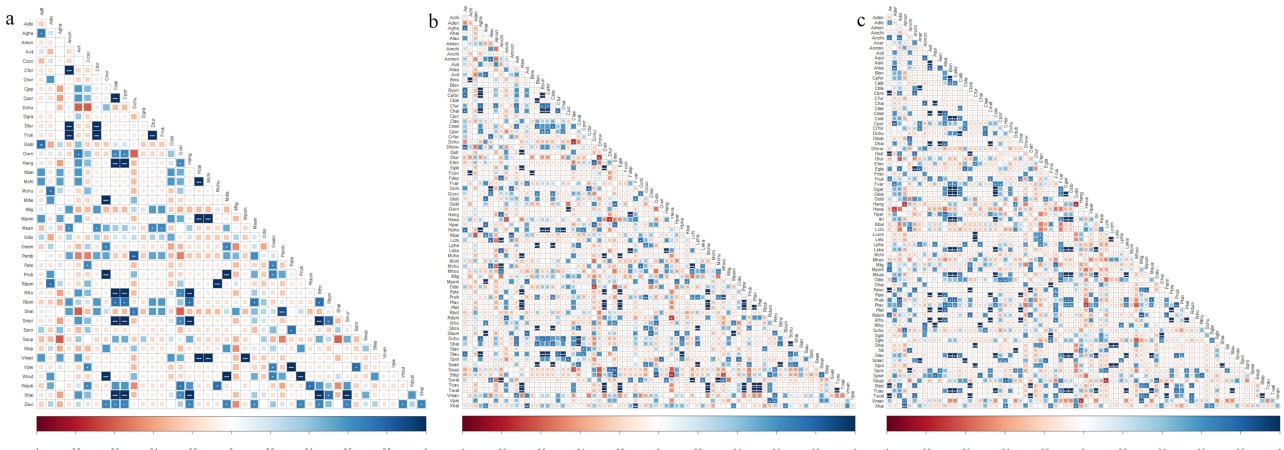

**Figure 3.** Spearman's rank correlation coefficients and significance of tree species in different successional stages. In the figure, (**a**) represents forests of early successional stages, (**b**) represents forests of early-mid successional stages, (**c**) represents forests of mid successional stages.

The results of Pearson's correlation and Spearman's rank correlation coefficients and significance indicated that the number of negative correlations of plants in the early-mid stage of succession was more than that in the mid stage of succession, which indicated that the competition of plant communities in the early-mid stage of succession was more obvious. This result was also the same as the result of niche overlap.

### 3.4. Soil Bacterial Diversity and the Influence of Successional Stage on Bacterial Community Structure

The observation and calculation results of the soil bacterial community (Figure 4) showed that there were significant differences in the species richness of soil microorganisms in the three successional stages, and there was no significant difference in chao1, PD tree, Shannon and Simpson diversity. Except for the richness, the other statistical results of the soil bacterial communities in the three successional stages were not significantly different, but they were nevertheless distinct in number. Chao1's trend change corresponded to richness's trend change, both of which were early-mid (993.77) > mid (992.94) > early (931.11). As the forest succession developed, PD whole tree became increasingly valuable, whereas Shannon became decreasingly valuable. Simpson's shifting tendency was similar in the early and mid stages (0.97), and it was greater than in the mid stage (0.96).

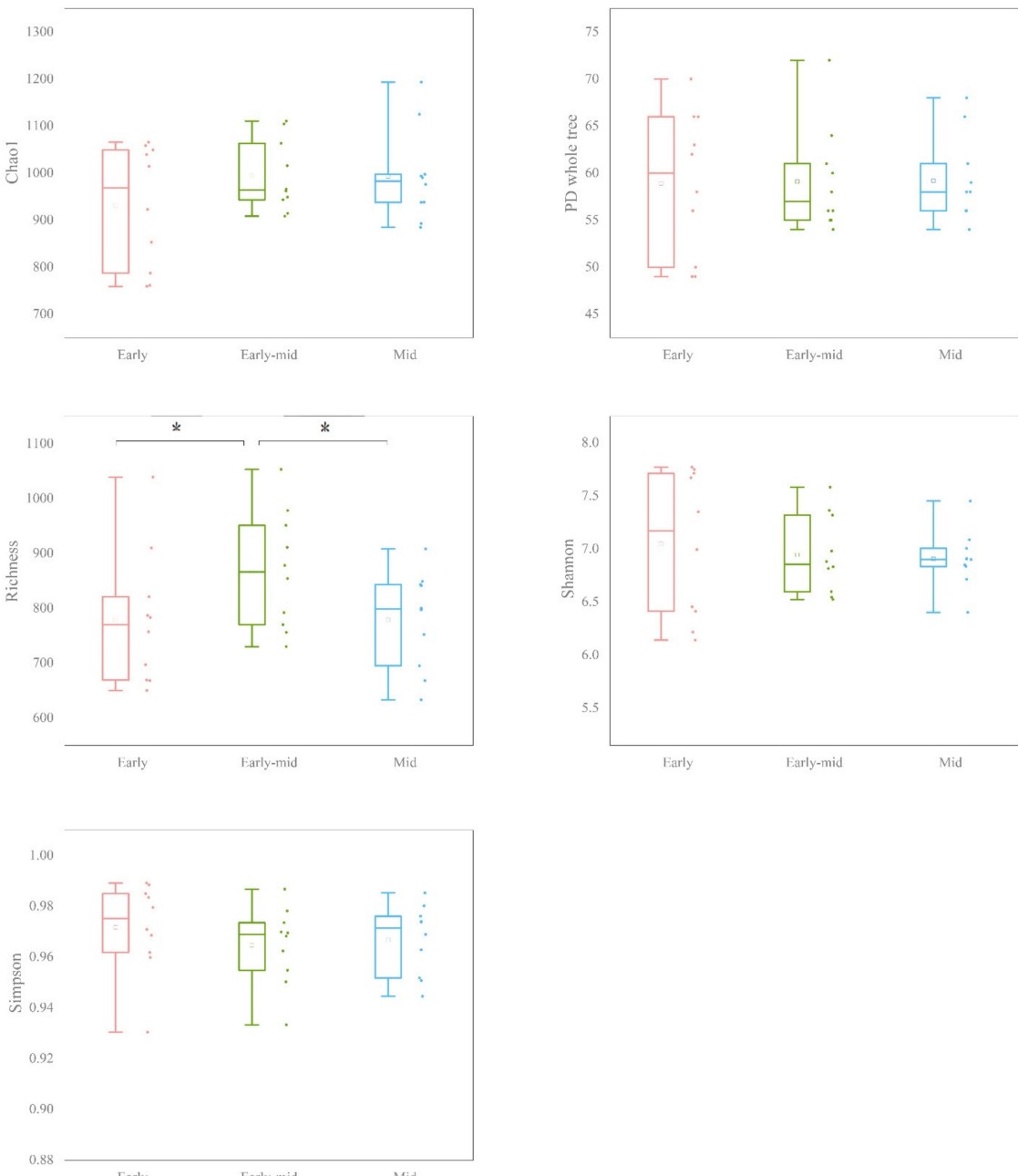

**Figure 4.** Bacterial Chao1, PD tree, richness, Shannon diversity and Simpson diversity comparison among different successional stages. * $p < 0.05$ by one-tailed *t*-test.

Richness is the number of species observed and chao1 is an index that uses an algorithm to estimate the number of sample OTUs. The trend changes of richness and chao1 were similar, and the statistical results of richness showed that the early-mid stage was significantly greater than the early and mid stages, indicating that the number of forest soil bacterial species in the early-mid stage of succession was greater than in the mid stage and early stage.

Distance-based redundancy analysis (db-RDA) was used to understand the potential effects of soil factors on soil bacterial composition at the phylum level (Figure 5). Collectively, in the early stages of succession, soil nutrient content had no discernible impact on the soil bacterial community, and the impact in the early-mid stage was stronger than in the mid stage. By comparing the calculation results in the early-mid and mid stages, it can be found that the P content, including the TP and AP content, had the greatest impact on the soil bacterial community. The N content, including TN and AN, had the deepest impact on the soil bacterial community.

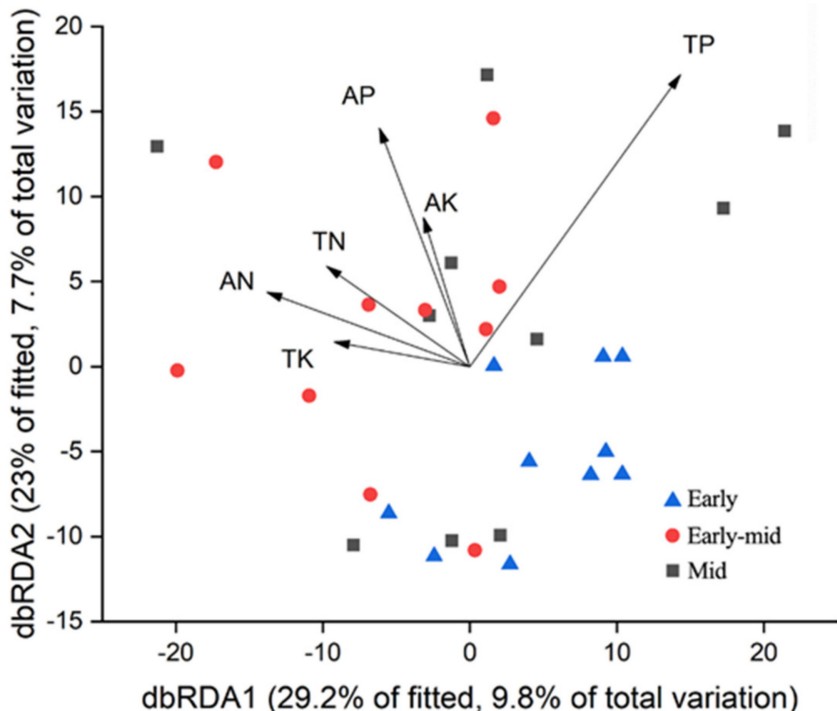

**Figure 5.** dbRDA ordination plot illustrates the environmental factors that best explain the variation of soil bacterial community at early, early-mid and mid restoration stages. AK: available potassium; AN: available nitrogen; AP: available phosphorous; TK: total potassium; TN: total nitrogen; TP: total phosphorus.

### 3.5. The Stochasticity of Soil Bacterial Communities

Normalized stochasticity ratio (NST), a new general mathematical framework to quantify ecological stochasticity under different situations, can serve as a better quantitative measure of stochasticity and was used to reflect the contribution of stochastic assembly among different successional stages. Results (Figure 6) showed that the stochasticity on the soil bacterial community decreased as the number of forest succession years increased. There was a significant difference between the early and mid successional stages.

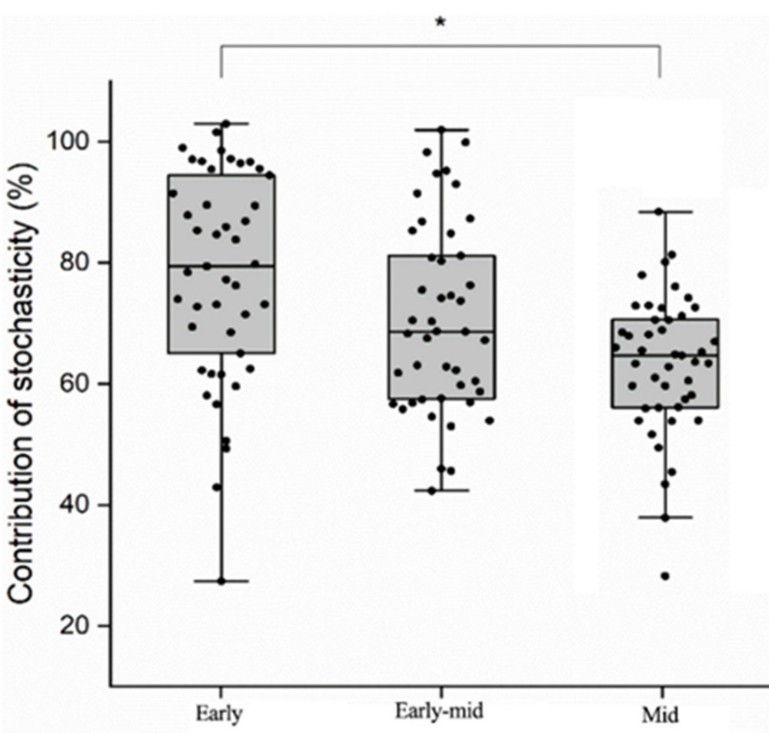

**Figure 6.** Contribution of stochasticity on the soil bacterial community assembly at early, early-mid and mid successional stages. * $p < 0.05$ by one-tailed *t*-test.

## 4. Discussion

### 4.1. The Development Direction of Plant Community Structure and Diversity

There were always two opposing opinions of succession in classical ecology [18], one is the classical view of succession proposed by E. P. Odum (1916), and the other is the simplification theory proposed by Gleason (1975). The classic concept of succession holds that community transformation is a sequential process that can be predicted; succession is the outcome of changes in the physical environment generated by the community, and succession is the culmination of its development with the formation of a top-level community. The content of the simplification theory is that succession is not orderly and predictable. The succession of plant communities is only the sum of individual dynamics.

Our research results showed that from the perspective of plant composition, the number of families, genera and species were increasing along with the progress of succession. Additionally, the changes in the ranking of important values along with the progress of the forest succession were consistent with the niche width calculated by the previous stage of the plant community (Table S1). This demonstrates that the properties of the previous stage's plant community can be used to forecast the composition of the plants in the next stage and that the predictions are credible. This finding is comparable to those found in previous investigations, stating that plant niches may be able to predict how plants replace themselves over time and space [21,22] and in the process of succession, the phylogeny is in a non-random mode [23–25]. This finding also provides tantalizing support on the classic view represented by E. P. Odum.

The result of niche overlap (Figure 1) showed that the overlap percentage in the early-mid stage was 41.90% and 37.56% in the mid stage, indicating that the early-mid stage competition is more intense. The results of Pearson's (Figure 2) and Spearman's rank correlation coefficients and significance concerning tree species in different (Figure 3) successional stages confirmed the same fact, that the competition of plant communities in the early-mid stage of succession was more obvious than in the mid stage. However, the changing trend of plant richness and diversity (Table S1) was larger in the mid stage than in the early-mid stage. This may be because there were many fierce competitions in the

early-mid stage, which caused some species to fail in the process of competing with other species, freeing up some ecological resources and giving other species a chance to develop more [26,27].

*4.2. Changes in Plant Communities, Soil Environmental Factors and Soil Bacterial Communities*

Plant–soil feedbacks are increasingly recognized as having a significant impact on the driving of plant communities and the succession process of terrestrial ecosystems [28–30]. There are numerous ways in which plants can influence the composition of soil communities [31]. Certain soil organisms are shown to create intimate bonds with specific plant species [32], and plant species identity could be a strong predictor of variation in soil communities [31].

In our study, the soil nutrient content of the tropical rain forest in the early stage of succession was found to be significantly less than the content in the early-mid and mid stages, whereas although there was no significant variation between the nutrient content in early-mid and mid stages, the early-mid stage's nutrient content was higher than the mid's (Table 1). Generally speaking, early successional ecosystems on forest sites are frequently distinct from those in the middle or later stages. The microclimate of a place is substantially altered when the overstory forest canopy is removed during disturbances. Increased sunshine exposure, more severe temperatures (ground and air), higher wind velocities and lower relative humidity and moisture levels in litter and surface soil are all consequences of these changes [33]. Shifts in these environmental metrics may change the plant species and soil nutrient content in the successional stage and make them considerably different from those of the forest after the canopy is restored (early-mid and mid successional stages). These factors help to explain the findings of our study that the soil nutrient content of the tropical rain forest in the early stage of succession was found to be significantly less than those in the early-mid and mid stages.

The leaching effect of rain may be one factor that affects the soil nutrient content. Strong precipitation may cause soil nutrients to be lost to low altitude areas in tropical rainforest zones. The early-mid successional forest had a lower average elevation than the mid stage (Table 1). This could be one of the reasons why the soil nutrient concentration in the early-mid stage was higher than in the mid stage. Furthermore, plant communities may be another significant influence on soil conditions. For a long time, people have known that some unique plants (e.g., Fabaceae species) can affect soil nutrient content [34,35]. We discovered that in the early-mid and mid successional stages of the tropical rainforest, there were two types of Fabaceae plant species in these stages, respectively (Table S1). These species were *Albizia attopeuensis* (the rank of IV was 42.11%, Niche breadth was 1.74) and *Peltophorum pterocarpum* (the rank of IV was 75.00%, Niche breadth was 1) (in the early-mid stage) and *A. attopeuensis* (the rank of IV was 35.44%, Niche breadth was 1) and *Sindora glabra* (the rank of IV was 100.00%, Niche breadth was 1) (in mid stage). Overall, Fabaceae plants had a higher ranking of important values and a wider niche width in the early-mid stages, indicating that they play a more vital role in the entire community. As a result, the amount of nitrogen stored in the soil rose (Table 1).

Significant differences were found in the species richness of soil microorganisms in the three successional stages. Meanwhile, their changing trends were consistent with the changes in soil nutrient content, that is to say, the value of the early stage < mid stage < early-mid stage (Figure 4). The result of db-RDA analysis also illustrated the impact of soil nutrient content in the early-mid stages was stronger than in the mid stages (Figure 5). Plant succession is essentially the interaction between aboveground plants and belowground microorganisms [36]; existing research proved that microbial community composition was affected by various factors such as land-use history [37] and C, N and P contents [5]. Soil nutrients such as N [38], ammonium, nitrate, available phosphorus and potassium [39] increase their availability when stimulated by plant root exudates. Furthermore, positive plant–microbe interactions such as the association with plants and plant-growth-promoting rhizobacteria (PGPRs) were already determined [40]. Therefore,

our research results showed that the nutrient content of soil had a positive and direct impact on the soil microbial community.

*4.3. Succession of Forest Plant Communities and the Stochastic of Soil Bacterial Community Establishment*

Understanding the processes and mechanisms behind the biodiversity patterns across space and time is one of the major goals in an ecological study focusing on community-scale [41–43]. Both deterministic and stochastic processes can control community assembly [20]. In our study, the contribution of stochasticity on the soil bacterial community formed a clear gradient along the restoration stage. The stochasticity of the soil bacterial community at the early successional stage of the rainforest was significantly higher than that at the mid stage (Figure 6). That is to say, the mechanism of soil bacterial colonization during the early successional stage tends to be more random, which typically includes random birth–death events, probabilistic dispersal (e.g., a random chance for colonization) and ecological drift (random changes in organism abundances) [41]. As the succession progresses, the construction of soil bacterial communities is more likely to be affected by deterministic processes from environmental factors. That is to say, community composition was initially governed by stochasticity, but as succession proceeded, there was a progressive increase in indeterministic selection correlated with environmental factors [44]. This further explains the result that the correlation between environmental factors and soil bacterial community during the early restoration stage was obviously less than that during the early-mid and mid successional stage (Figure 5).

## 5. Conclusions

The diversity of plant communities expanded as the succession of tropical lowland rainforests in Hainan developed, but the competitive interaction between communities decreased, and the general status of the forest became more stable. Specifically, the soil nutrient content increased in the early to early-mid stage and decreased in the early-mid to mid stage. This may be related to the phenomenon that we have observed more Fabaceae plants in the early-mid stage. In the process of forest succession, changes in soil environmental factors were positively affecting the changes of soil bacterial communities, and with the progress of forest succession, the stochasticity of soil bacterial communities was decreasing lower and lower. Overall, as a result of forest succession, the diversity of plant communities increased, the competition and stochasticity of soil bacterial communities decreased and soil nutrient content changed.

**Supplementary Materials:** The following supporting information can be downloaded at: https://www.mdpi.com/article/10.3390/f13020348/s1, Table S1a: The niche breadth and IV ranking of tree species in early successional stage; Table S1b: The niche breadth and IV ranking of tree species in early-mid successional stage; Table S1c: The niche breadth and IV ranking of tree species in mid successional stage.

**Author Contributions:** X.H. analyzed the data and wrote the paper. Q.S., Z.S. and W.G. conducted the field investigation and sample analyses, L.Q. contributed to the draft manuscript. All authors have read and agreed to the published version of the manuscript.

**Funding:** This work was jointly funded by the Fundamental Research Funds for ICBR (1632019006, 1632021023) and the National Key R&D Program of China (2021YFD2200405).

**Data Availability Statement:** Not applicable.

**Acknowledgments:** This work was jointly funded by the Fundamental Research Funds for ICBR (1632019006, 1632021023) and the National Key R&D Program of China (2021YFD2200405). Thanks to Junwei Luan and Ruijing Xu for their help in the process of conducting experiments and writing the manuscript.

**Conflicts of Interest:** The authors declare no conflict of interest.

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
