# Peer review of "Secondary Succession in the Tropical Lowland Rainforest Reduced the Stochasticity of Soil Bacterial Communities through the Stability of Plant Communities"

_forests, doi:10.3390/f13020348_

Round 1

Reviewer 1 Report

Line 62: should be in italics

Line 73: it is better to talk about forests of different ages or give the classification according to the literature

Lines 87-97: I am not sure if we can tell about stages of succession: early and mid. Perhaps it is better to talk about the forest of different ages

Lines 184, 187, 192, Table 1(line 194): as above

Line 199: as above, rather forests in different ages

Lines: 202, 203 as above

Lines 213-217 as above

Line 227: may be earlier in succession

Line 229: later in succession

Line 235: as above

Table 2a, b, c : as above

In section 3.3. Plant niche overlap and interspecific associations- as above, terms: early, early-mid and mid please change to  forests different ages, and in other sections of the text

Author Response

Thanks for your helpful comment, it helps a lot. Based on your suggestion, we have revised the article accordingly. We apologize for not listing enough references and definitions in the “Study site description” section to make you confused. Please see the revised manuscript for details.

Thank you for your kind help in the publication of our article!

For the point-by-point response please see the attachment.

Have a great day!

Reviewer 2 Report

In this study, the authors investigated the dynamics of plant communities, soil properties and the structure of soil bacterial communities in the tropical forests representing 33, 60 and 73 years, respectively. The primary objective was to explore the influence of forest succession on those mentioned parameters. However, there are several shortcomings for this manuscript. For example, the concept “successional stage” and “restoration stage” is misused. In the Introduction, the authors mentioned that importance of endemic plant species, why? In the biomes of a certain region, there are always exist endemic species. Overall, the manuscript is in poor description.  

  1. The title of the manuscript is too long and a little confused to readers.
  1. In the Abstract, the authors should point out the indication of “early-mid” and “mid”.
  2. Line 25-27, the sentence is very difficult to understand.
  3. Changed “succession stage” to “successional stage” throughout the manuscript.
  4. Line 28, changed “in” to “among”.
  5. Line 29, the authors should describe the variables under the effect of soil nutrient content.
  6. Overall, the Abstract is in poor description. It is strongly recommended to reorganize it.
  7. Line 42-46, the sentence should be reorganized for clear.
  8. Line 58, delete “has”.
  9. Line 62, the name of the plant species should be italic.
  10. Line 74-75, the sentence should be reorganized for clear.
  11. In the “study site description”, the authors selected a natural successional chronosequence of the rainforest, and defined different successional stages (i.e., 33-year-old forest as early successional stage, 60-year-old forest as early-mid successional stage, 73-year-old forest as mid successional stage). What’s the reference for this definition? The age of the late two kind of forests is very close.
  12. Line 102-103, the sentence is not clear. What does restoration stage refer to? Do you mean the plots within each successional stage were separated each other by at least 500m?
  13. Line 104, what does quadrat refer to?
  14. Line 105-108, the sentence should be reorganized for clear.
  15. The method in sampling soil samples should be described clearly. How many soil cores in each plot? As you said, there were 30 soil samples in total, however, you collected the soil samples in three depth.
  16. The references for determining soil properties should be added.
  17. Line 165-166, I suggested the authors to describe the methods for calculating important vale ha niche breadth in details.
  18. Added “soil” before “bacterial”.
  19. Line 190, change “at” to “among”.
  20. Line 191-192, the sentence is difficult to understand.
  21. Line 216-220, I strongly suggest that placing these description in the Discussion section.
  22. In Table 2, the full name of IV should be listed. Additionally, I suggested the authors change the Table 2 as supplementary materials.
  23. Most of the Figures in the main text are in low quality and needed to be improved.
  24. In the 4.3 sub-section of Discussion, the two paragraphs could be merged. The discussion on the mechanisms underlying forest succession effects on the stochastic of soil bacterial community is not enough.
  25. The Conclusion should be reorganized and focused on the findings. For example, the sentence in line 420-421 could be deleted.

Author Response

Thanks for your helpful comment, it helps a lot. Based on your suggestion, we have revised the article accordingly. Please see the revised manuscript for details.

Thank you for your kind help in the publication of our article!

For the point-by-point response please see the attachment.

Have a great day!

Reviewer 3 Report

The review of your manuscript was sent to Editor Office by e-mailed .

Dear Authors,

I have some comments and suggestions to your manuscript, following:

  • Title of the article: Why is the title written in the form of conclusions?

The title should be shorter and lighter.

  • Line 125:

 The Kjeldah method is used to determine total nitrogen (it is the sum of organic -N and compounds of mineral nitrogen, ammonia and ammonium), therefore please explain the difference between total nitrogen and alkyl nitrogen estimated by the Kjeldahl method?

  • Line 126:

 There  is not „Kjeldahl 2003, Foss, Denmark” in Reading Chapter of manuscript.

  • 4.

There are no asterisks in the sub-pictures of fig. 4, while the asterisk is explained in the legend.

  • Why have you formed subject of 4.2. subchapter as a conclusion of results?

„4.2. Changes in plant communities affect soil environmental factors, which in turn positively    affect soil bacterial communities”,

This chapter should be supplemented with knowledge about the significant impact of plant root exudate on the abundance as well as the taxonomic and  physiological diversity of soil microorganisms.

  • Please correct the shape of the title in chapter 4.1, 4.2. 4.3 as it is the same situation at the beginning.

Title 4.1.; 4.2. and 4.3. are presented as a conclusion.

Author Response

Thanks for your helpful comment, it helps a lot. Based on your suggestion, we have revised the article accordingly.

Thank you for your kind help in the publication of our article!

For the point-by-point response please see the attachment.

Have a great day!

Round 2

Reviewer 2 Report

Line 32-33, the sentence “xxx the impact of soil nutrient content effects of soil factors xxx ”should be revised.

Line 215, changed “at” to “among”.

Line 473, “successional stage” could be deleted.

Author Response

Thanks for your careful and helpful comment. Please see the revised manuscript for details.

Thank you for your kind help in the publication of our article!

Have a good day!
